# In-Vitro, In-Vivo, Molecular Docking and ADMET Studies of 2-Substituted 3,7-Dihydroxy-4H-chromen-4-one for Oxidative Stress, Inflammation and Alzheimer’s Disease

**DOI:** 10.3390/metabo12111055

**Published:** 2022-11-02

**Authors:** Mater H. Mahnashi, Mohammed Abdulrahman Alshahrani, Mohammed H. Nahari, Syed Shams ul Hassan, Muhammad Saeed Jan, Muhammad Ayaz, Farhat Ullah, Osama M. Alshehri, Mohammad Ali Alshehri, Umer Rashid, Abdul Sadiq

**Affiliations:** 1Department of Pharmaceutical Chemistry, College of Pharmacy, Najran University, Najran 61441, Saudi Arabia; 2Department of Clinical Laboratory Sciences, Faculty of Applied Medical Sciences, Najran University, Najran 61441, Saudi Arabia; 3Shanghai Key Laboratory for Molecular Engineering of Chiral Drugs, School of Pharmacy, Shanghai Jiao Tong University, Shanghai 200240, China; 4Department of Natural Product Chemistry, School of Pharmacy, Shanghai Jiao Tong University, Shanghai 200240, China; 5Department of Pharmacy, Bacha Khan University, Charsadda 24420, KP, Pakistan; 6Department of Pharmacy, Faculty of Biological Sciences, University of Malakand, Dir (L), Chakdara 18000, KP, Pakistan; 7Medical Genetics Laboratory Sciences, Faculty of Applied Medical Sciences, Najran University, Najran 61441, Saudi Arabia; 8Department of Chemistry, COMSATS University Islamabad, Abbottabad Campus, Abbottabad 22060, KP, Pakistan

**Keywords:** flavone, *Notholirion thomsonianum*, cholinesterases, oxidative stress, inflammation

## Abstract

Plants’ bioactives are well-known safe drugs for vital diseases. Flavones and Flavonoid-rich dietary supplements are known to exhibit neuroprotective potential. In this study, we isolated a flavone 2-(3,4-dimethoxyphenyl)-3,7-dihydroxy-4H-chromen-4-one from *Notholirion thomsonianum* and it was evaluated against various targets of the oxidative stress-related neurological disorders. The compound showed excellent acetyl and butyrylcholinesterase inhibitions in its profile, giving IC_50_ values of 1.37 and 0.95 μM, respectively. Similarly, in in-vitro MAO-B assay, our flavone exhibited an IC_50_ value of 0.14 μM in comparison to the standard safinamide (IC_50_ 0.025 μM). In in-vitro anti-inflammatory assay, our isolated compound exhibited IC_50_ values of 7.09, 0.38 and 0.84 μM against COX-1, COX-2 and 5-LOX, respectively. The COX-2 selectivity (SI) of the compound was 18.70. The compound was found safe in animals and was very effective in carrageenan-induced inflammation. Due to the polar groups in the structure, a very excellent antioxidant profile was observed in both in-vitro and in-vivo models. The compound was docked into the target proteins of the respective activities and the binding energies confirmed the potency of our compound. Furthermore, absorption, distribution, metabolism, excretion, and toxicity (ADMET) results showed that the isolated flavone has a good GIT absorption ability and comes with no hepatic and cardiotoxicity. In addition, the skin sensitization test, in-vitro human cell line activation test (h-CLAT) and KeratinoSens have revealed that isolated flavone is not skin sensitive with a confidence score of 59.6% and 91.6%. Herein, we have isolated a natural flavone with an effective profile against Alzheimer’s, inflammation and oxidative stress. The exploration of this natural flavone will provide a baseline for future research in the field of drug development.

## 1. Introduction

Alzheimer’s disease (AD) is a progressive age-related neurodegenerative disorder which is highly prevalent among individuals above the age of sixty [1]. AD is a major cause of dementia and is typically associated with cognitive decline, behavioral turbulence, and imperfection in routine activities [2,3]. It is highly prevalent among the elder population and affects 45 million people globally, which is expected to triple by 2050 [4]. The deficiency of the acetylcholine neurotransmitter is among the major pathological targets of AD. Besides the ACh neurotransmitter, deposition of amyloid plaques and neurofibrillary tangles are also important. Moreover, free radicals also cause mitochondrial dysfunctions and lead to inflammation and neuronal degeneration [5,6,7]. Among the current treatment strategies include: use of ACh metabolizing enzymes (cholinesterase’s inhibitors), anti-amyloid therapeutics, antioxidant and anti-neuroinflammatory agents. Unfortunately, the drug discovery against AD is very limited. Only four drugs are clinically approved, among which three are cholinesterase inhibitors (galantamine, donepezil, rivastigmine), whereas the fourth memantine is a glutamate regulator [8,9]. These drugs have limited efficacy and are effective for symptomatic relief only. Among these drugs, galantamine is isolated from natural products and rivastigmine is a synthetic analogue of naturally occurring compound [7,10,11].

Free radicals are generated during normal metabolic process and are scavenged by the immune system’s antioxidant system [12]. However, excessive liberation of radicals leads to oxidative stress. Oxidative stress refers to an imbalance between the fabrication of free radicals and its neutralization by the body’s protective mechanisms [13]. The free radicals generated thus cause oxidation of biological molecules and structural alternation in proteins, as well as causing mutations in the gens which trigger signaling cascades, thus initiating inflammatory responses or progressing these processes further. Thus oxidative stress causes the activation of several transcription factors which ultimately causes the expression of pro-inflammatory genes, which activate inflammatory pathways [14]. Free radicals trigger oxidative stress, thus lead to various chronic inflammatory disorders [15]. For instance, in Alzheimer’s disease (AD), excessive free radicals liberated by mitochondrial poison Aβ cause disruption of metabolic processes, including energy production, and attack neuronal cells leading to their death [16]. In situations where there is excessive free radical production, or the immune system is compromised, then supplementation of exogenous antioxidants is extremely necessary. 

Flavonoids and flavonoid-rich dietary supplements are known to protect against degenerative disorders of nervous systems and promote healthy aging [17,18]. These agents are known to improve cognitive performance, suppress neuro-inflammatory pathways, scavenge free radicals, inhibit cholinesterases and exhibit neuroprotective potential [19]. Flavonoids are known to modulate signaling pathways implicated in synaptic plasticity, reduce neuroinflammation, stimulate neuronal cell growth and reduce the accumulation of pathological proteins [20]. Flavonoids mediate their neuroprotective effects via modulation of important signaling pathways which cause inhibition of neurotoxin-mediated neuronal cells apoptosis [21]. Furthermore, they promote neuronal survival and differentiation by regulating pathways including PI3/Akt and MAP kinase which in turn regulate the expression of genes and pro-survival transcription factors [19,22]. Furthermore, they improve cerebral blood flow, induce angiogenesis, improve nerve cell growth and augment neurotrophin release and function [23]. Subsequently, flavonoids and flavonoid-rich products, including natural products and nutraceuticals, prevent abnormal aging, inhibit neuronal degeneration and improve cognitive performance [24,25].

*Notholirion thomsnianum* (Royale) Stapf, a member of the Liliaceae family, is traditionally used to treat infectious diseases, gastrointestinal disturbances and to treat tumors [26]. We previously validate the plant for its analgesic, antimicrobial and anti-diabetic potential [27,28,29]. The aim of our designed work was to isolate a pure compound (2-(3,4-dimethoxyphenyl)-3,7-dihydroxy-4H-chromen-4-one) from the potent fraction of *N. thomsonianum*. The isolated compound was subjected to in-vitro cholinesterase inhibitory, MAO-B inhibitory, antioxidant and anti-inflammatory studies. After the in-vitro results, the isolated compound was tested safely in experimental animals and was then subjected to in-vivo anti-inflammatory assay. Furthermore, the isolated compound was evaluated for its effects on in-vivo antioxidant markers including Superoxide Dismutase (SOD), Catalase (CAT) and lipid peroxidation. The molecular docking and ADMET studies were also performed to strengthen our design work.

## 2. Materials and Methods

### 2.1. Plant Sample Collection and Extraction

Our selected medicinal plant was collected from District Swat Pakistan after prior authentication by a plant taxonomist, Dr. Nasrullah Khan, from the Department of Botany, University of Malakand, Chakdara, Pakistan. Subsequently, a dried sample of the plant was deposited at the herbarium of the same university with reference voucher no. (H.UOM.BG.106) for future reference. Plant materials were shade dried, converted to powder for using in a cutter mill and were subsequently processed for extraction and fractionation. Powder plant materials (500 g) were macerated in 1000 mL of 80% methanol at room temperature for about two weeks with periodic shaking to get the constituents completely dissolved. Subsequently, the solvent was filtered via Whatman paper 1, and solvent was evaporated via rotary evaporator (Heidolph Laborota 4000, Schwabach, Germany) to get crude extract (400 g) [30].

### 2.2. Isolation and Identification of Compound

Initially, we performed the TLC analysis of crude extract using n-hexane and ethyl acetate in different ratios. Subsequently, we also tested combinations of other solvent systems like chloroform, methanol and dichloromethane. Based on the initial TLC analysis, we selected n-hexane and ethyl acetate as an optimal solvent system to elute the column. We started the column with low polarity and gradually increased the polarity by adding ethyl acetate to n-hexane. The starting ratio of n-hexane to ethyl acetate was 100:0 which was gradually increased by a five-digit difference (based on the in-process TLC analysis). The eluting ratio was 50:50 of both of the solvents. We then randomly checked different eluted fractions using TLC analysis. The eluted fractions with a common retardation factor on TLC were combined and were dried using reduced pressure rotary evaporator. The purified compounds with some unknown impurities were then loaded onto a relatively smaller silica gel packed column for further purification. Finally, we were able to get a pure single spot on the TLC. The compound’s structure was confirmed with ^1^H NMR, ^13^C NMR and MS analyses [31,32].

### 2.3. In-Vitro Anti-AD Assays

#### 2.3.1. Anti-Cholinesterase Assays

For the determination of anti-cholinesterase activity of the compounds, the acetylcholinesterase enzymes used were obtained from an electric eel. The butyrylcholinesterase (BChE) is obtained from equine serum. The principle of this assay is based on the hydrolysis of acetylcholine iodide by the enzymes AChE and the hydrolysis of butyrylcholine iodide by the enzyme butyrylcholinesterase. This process of hydrolysis yields 5-thio-2-nitrobenzoate anion which then form a complex with DTNB (yellow color) which can be detected by spectrophotometer [33].

##### Preparation of Solution

We dissolved the compound separately in a phosphate buffer (0.1 M) by forming different concentrations that ranged from 250–1000 mcg/mL. Initially, KH_2_PO_4_ (13.6 g/L) and K_2_HPO_4_ (17.4 g/L) were mixed in a ratio of 6% and 94% respectively for the preparation of 0.1 M of phosphate buffer solution with a pH of 8.0 ± 0.1. Then KOH was used for the adjustment of pH. Then, we diluted AchE (solid 518 u/mg) and BChE (7–16 U/mg) in freshly prepared phosphate buffer until we obtained the final concentration of 0.03 U/mL and 0.01 U/mL. Afterwards, we prepared the solutions of acetylcholine (0.0005 M), BTchI (0.0005 M) and DTNB (0.0002273 M) in distilled water and kept it in an Eppendorf cap and placed it in a refrigerator. We dissolved galantamine in methanol which was used as a positive control, and then prepared further dilutions as previously mentioned [33].

##### Spectroscopic Analysis

In each of these enzyme assays, the enzyme solution (5 μL) was added to the cuvette followed by the addition of the compound (205 μL), and finally DTNB reagents (5 μL) were added. The solution was maintained in a water bath at a temperature of 30 °C for 15 min, and then the substrate solution (5 μL) was added. The absorbance was measured at 412 nm by using a double beam spectrophotometer. The negative control consists of all the components except the compound. Galantamine (10 μg/mL) was used as a positive control in the assay which is a standard cholinesterase inhibitor. The absorbance, as well as the reaction time, were taken for 4 min at a temperature of 30 °C and were repeated in triplicate [33].

### 2.4. MAO-B Inhibition Assay

Human recombinant MAO-B enzymes used in this assay were obtained from insect cells. The substrate used for MAO-B enzyme is benzylamine hydrochloride. For all the enzymes reactions, as well as for dilutions sodium phosphate buffer (0.05 M), a PH of 7.4 is used. Finally, a volume of 200 μL of enzyme reaction was used in each reaction containing every substrate, different concentrations of the test inhibitors (0.01 μM to 100 μM) and DMSO 1% was used as a cosolvent. The activity of the test compound hMAO-B assay is determined by calculating its effect on H_2_O_2_ production from benzylamine while using an Amplex Red MAO assay kit (Invitrogen). In this assay, a volume of 0.1 mL sodium phosphate buffer (0.05 M) with a PH of 7.4, which contains different concentrations of the test compound (reference inhibitors or new compound) and a sufficient quantity of recombinant hMAO-B (0.5 μL, 71 Unit/mg) is incubated at 37 °C for 1 h in a 96-well plate. After incubating, the reaction was initiated by the addition of (final concentration) 200 μM Amplex Red reagent, 1 mM benzylamine hydrochloride and 1 Unit/mL horseradish peroxidase. The H_2_O_2_ production, and finally resorufin, was measured in a multi-detection microplate fluorescence reader (TECAN) at 37 °C and with an absorbance of 570 nm. Simultaneously, the control experiment was also carried out by simply testing the compound with suitable vehicle dilutions. Similarly, the activity of the given test compound to modify through nonenzymatic inhibition (for direct reaction with Amplex Red reagent) was found by the addition of these compounds to the solution which contains Amplex Red reagents only in buffer. Then, specific fluorescence absorbance was determined after subtracting the background activity, measured from the vials which contains all the components except hMAO-B, and was replaced by buffer solution [34].

### 2.5. In-Vitro Antioxidant Assays

#### 2.5.1. DPPH Free Radical Scavenging Assay

Free radicals and 1,1-diphenyl, 2-picrylhydrazyl (DPPH) were used for the illustration of the free radical scavenging ability of the molecule [35]. This assay involved the addition of different dilutions (250, 500 and 1000 μg/mL) of this compound to 0.004% methanolic solution of DPPH. The absorbance was measured at 517 nm on a UV Spectrophotometer after 30 min. Ascorbic acid, in this case, was used as a positive control. The percent activity was calculated according to the formula.

#### 2.5.2. ABTS Free Radical Scavenging Assay

For confirmation of antioxidant activity of a compound, the ABTS (2, 2-azinobis [3-ethylbenzthiazoline]-6-sulfonic acid) free radicals were used. The basic principle of this assay is the antioxidants’ ability to scavenge the ABTS and radical cation which reduces the absorbance at 734 nm. We prepared K_2_S_2_O_4_ (2.45 mM) and ABTS (7 mM) solutions and mixed them. We thne kept the final mixture in a dark place at room temperature for 12–16 h to obtain a dark colored solution which contained radical cations and ABTS. While determining the activity, we diluted the radical cation and ABTS solution with a phosphate buffer (0.01 M) with PH 7.4 and then at 734 nm adjust the absorbance value to 0.70. To determine the radical scavenging activity of the compound, we added 3.0 mL of ABTS solution to 300 mL of the compound in a cuvette. By spectrophotometer, we measured the absorbance reduction one minute after the mixing of the solution and continued measuring it for 6 min. For positive control, ascorbic acid is used. We then repeated the assay three times and calculated the percent of inhibition according to the formula [35].

### 2.6. In-Vitro Anti-Inflammatory Assays

#### 2.6.1. Anti-Cyclooxygenases Assays (COX-1 and COX-2)

We activated the enzyme Cyclooxygenase-1 (COX-1) (10 mL, 0–0.7–0.8 mg and Cyclooxygenase -2 (COX-2) (300 U/mL) on ice for 5 min along with adding 50 mL of cofactor solution, which contained 0.1 mM hematin, 0.9 mM glutathione and 0.24 mM TMPD (N,N,N,Ntetramethyl-p-phenylenediamine dihydrochloride) in 0.1 M Tris HCl buffer with PH 8.0. We then placed 20 mL of the test sample with different concentrations ranging from 31.25–1000 mg/mL with enzyme solution (60 mL) for 5 min at room temperature. By adding 20 mL arachidonic acid (30 mM) the reaction started. After that reaction, the mixture was incubated for 5 min. By using UV-visible spectrophotometer, at 570 nm the absorbance was measured. Then, from absorbance value per unit time, the % inhibition of COX-1 and COX-2 enzymes were calculated. By plotting the sample solution concentration against inhibition, the IC_50_ value was determined. Mostly indomethacin was used as a positive control for COX-1 while celecoxib was used as a positive control for COX-2 [36,37].

#### 2.6.2. 5-Lipoxygenase Inhibitory (5-LOX) Assay

Using an established procedure, the 5-lipoxygenase activity was performed. Firstly, we prepared various concentrations of test compound ranging from 31.25–1000 mg/mL. Afterwards, the 5-lipoxygenase solution (10,000 Unit/mL) was prepared. Linoleic acid (80 Mm) was the substrate used for the 5-LOX assay. We also prepared phosphate buffer (50 Mm) with PH 6.3. WE dissolved the various concentrations of the compound in each 0.25 mL of phosphate buffer, then added 0.25 mL of lipoxygenase enzyme to it and incubated it at 25 °C for 5 min. Then, we added 1 mL of linoleic acid solution (0.6 Mm), mixed it and measured the absorbance at 234 nm. We repeated this experiment three times. The standard drug used was Zileuton [36]. By using the following equation, percent inhibition was calculated:$$\mathrm{Percent}\;\mathrm{inhibition}=\frac{\mathrm{Absorbance}\;\mathrm{of}\;\mathrm{control}-\mathrm{absorbance}\;\mathrm{of}\;\mathrm{sample}}{\mathrm{Absorbance}\;\mathrm{of}\;\mathrm{control}}\times 100$$

### 2.7. In-Vivo Studies

#### 2.7.1. Experimental Animals and Acute Toxicity

The Albino mice with an average body weight of 25–30 g were obtained from NIH, Islamabad, Pakistan from our designed studies. The animals were kept in the animal house of the university and were kept under controlled conditions as per the standard guidelines. The animals were used in experiments according to the standard protocols as per the approval of Departmental Ethical Committee with approval letter number DREC-140 [36].

The animals were divided into various groups, each having five animals in a group. The tested sample was given in different groups at the dose of 25, 50, 100, 200, 500, 1000 and 2000 mg/kg b. wt. After dose administration, the tested animals were monitored carefully for the next 3 days for any kind of spontaneous activity, aggressiveness, cyanosis, ataxia, tail pinch reaction, righting reflex, writhing, allergic indication, catalepsy convulsions, and any other abnormal performance.

#### 2.7.2. In-Vivo Carrageenan Method of Inflammation

The progression of inflammation stimulated by carrageenan is a biphasic system. In the initial phase the serotonin and histamine were released, while in the second phase of swelling prostaglandin, bradykinin, leukotriene and lysozyme were mediated. The non-steroidal and steroidal drugs can wield their function in the second phase of the edema formation. The edema inhibition of the tested compound was clearly seen in the second phase and that could be due to the inhibition of prostaglandin secretion. Along with this, other possible mechanisms of action of the tested sample was by the histamine suppression, bradykinin, which is concerned in the initial and late phase of edema induced by carrageenan. The tested sample was originally more efficient in plummeting the carrageenan-induced paw edema and comparable to that of the standard drug [36].

Based on the encouraging in-vitro results, we subjected our isolated flavone to the animals based on anti-inflammatory studies using a standard carrageenan method [36]. In the assay, animals were randomly divided into three groups with eight in each group. The first group, Group I, was the negative control and was given dimethyformamide 1% 10 mL/kg bw. The second group (Group II) was the positive control and was given 100 mg/kg bw of aspirin. The third group (Group III) was given 10 mg/kg bw of the isolated flavone. After ½ h, 1% of the saline solution of carrageenan (*w*/*v*, 0.05 mL) was given in a sub-planter area of the animals. Afterwards, the paw edema was determined with the help of a plethysmometer (Model LE 7500 plan. Lab SL) with different time intervals. The paw edema of the mice administered with flavone, and with those of the standard drug, were measured at different time intervals. The measurements of paw edema were compared with the mice serving as the negative control group. The percent inhibition activity was calculated as per the standard procedure [36].

#### 2.7.3. Antioxidant Markers/In-Vivo

In this assay, the experimental animals were divided into four groups with five animals in each group. Group I was given the corn oil as a vehicle. The remaining three groups (Group I, II and III) were given CCl_4_ in corn oil in a ratio of 1 to 5 (*v*/*v*) for the induction of oxidative stress. The two groups were given 5 and 10 mg/kg bw of the isolated flavone for two weeks. One day after the last administration, the animals were given IM zoletil50 (30 mg/kg bw) with xylazine (5 mg/kg bw) for about 10 min to get anesthetized. Afterwards, the blood sample was taken out by cardiac puncture into EDTA-tubes. After the experiments, the animals were euthanized as per the standard procedure. The blood (200 μL) was centrifuged for 10 min at the rate of 3000 rpm. The plasma was used for the biochemical studies. Afterwards, 0.9% of physiological saline was added to the cellular layer to double up the volume. Again, it was centrifuged under the same set of conditions and the upper layer was discarded. The experiments were repeated three times and erythrocytes were isolated. The isolated erythrocytes were stored at a low temperature (−20 °C). The vital organs like kidneys, liver and heart were harvested, washed and weighted.

##### Analyses of Oxidative Stress Markers

The RBCs were employed in a cellular model to evaluate the potency of antioxidants to penetrate into the living cells from plasma.

Superoxide Dismutase Assay

The superoxide dismutase assay was performed as per the reported procedure. A sample of haemolysate (0.2 mL) was added to carbonate buffer (2.5 mL, 0.05 M having pH of 10.2). The reaction was initiated with the addition of adrenaline (0.3 mM, 0.3 mL) to the buffered mixture. This was mixed immediately and was subjected to a spectrophotometer as a sample cuvette. The solution of buffer (2.5 mL), substrate (0.3 mL) and distilled water (0.2 mL) were added in the reference cuvette. The increase in absorbance (480 nm) was monitored for 2.5 min with 30 sec intervals against a blank. The SOD was calculated as per the reported procedure [38].

2.Catalase Assay

The catalase assay on the isolated compound was performed as per the standard protocols. Six test tubes were taken, containing increasing concentrations of H_2_O_2_ (up to 640 μM). To each of these test tubes, potassium dichromate was added. The tubes were heated for 10 min up to 100 °C. The tubes were then allowed to cool down and optical density was measured at 570 nm. The standard curve of concentration vs. absorbance of H_2_O_2_ was plotted. Afterwards, haemolysate (1 mL) was diluted and added to the test tube containing H_2_O_2_ (2 mL) and phosphate buffer (2.5 mL, 0.01 M having pH 7.0) and were mixed. After every 30 sec, one ml of solution was withdrawn and was transferred to acidified potassium dichromate (2 mL) for 2 min. This was mixed and heated up to 100 °C for 10 min. The solution was then cooled down and optical density was measured (at 570 nm) against the blank. The catalase activity was calculated as per the standard method [39].

3.Lipid Peroxidation Assay

The lipid peroxidation was performed with malondialdehyde assay as per the reported methods. The diluted haemolysate (0.4 mL, 1:10 *v*/*v*) was added to the test tube containing acetic acid sodium hydroxide along with thiobarbituric acid, having a volume of 2 mL each. The tubes were heated for 20 min at 100 °C and absorbance was measured at 532 nm against the blank. The activity was then calculated as per the standard method [40].

4.Total Proteins Assay

The total proteins assay of the experiment was performed as per the standard method [39]. The diluted haemolysate (0.2 mL), bovine albumin (0.2 mL, 2 g/dl), distilled water (2 mL) and working reagent 1 mL (CuSO_4_ with NaK one part and Na_2_CO_3_ with NaOH 100 parts) were added in different test tubes. This was mixed and incubated at room temperature for 10 min. Then, Folin Ciocalteu Reagent (FCR) (0.25 μM, 50%) was mixed in and further incubated at normal laboratory temperature for 10 min. The absorbance was measured at 570 nm against the blank. The concentration of total proteins was determined as per the standard method [40].

### 2.8. Molecular Docking Studies

The molecular targets were further studied for their interaction exploration via docking studies. Docking studies were carried out by using Molecular Operating Environment (MOE) software [41,42]. First, we downloaded the three-dimensional crystal structure of the selected molecular targets from a protein data bank (PDB). The PDB accession code of the downloaded enzymes are: 1EVE for AChE, 4BDS for BChE, 1CX2 for COX-2, 6N2W for 5-LOX, 2V5Z for MAO-2 and 5I38 for DPPH. The preparation of the 3-D structure of ligand, downloaded enzymes and docking runs were carried out by our previously reported methods [36]. The interaction plot analyses were carried out by using Discovery studio Visualizer.

### 2.9. ADMET Analysis

ADMET (absorption, distribution, metabolism, excretion, and toxicity) are the essential measurement tools for any compound before being elected as a drug candidate. The online web tool swiss ADME (http://www.swissadme.ch/index.php accessed on 6 September 2022) was used to obtain the ADME properties of the isolated flavone [43] and the pharmacokinetic scores were predicted using the online web application pkCSM (http://biosig.unimelb.edu.au/pkcsm/prediction accessed on 6 September 2022)

### 2.10. Caridac Toxicicty

The blockage of the hERG channels are linked to the fatal cardiac arrhythmias. The web tool pre-hERG 4.2 (http://predherg.labmol.com.br/predict accessed on 6 September 2022) was used for early predictive cardiac toxicity.

### 2.11. Skin Sensitization

The skin sensitization of the isolated flavone was conducted through in-vitro cellular response (http://predskin.labmol.com.br/predict accessed on 6 September 2022), a web tool used for early predictive skin sensitization [42].

### 2.12. Statistical Analysis

The One-way ANNOVA test was used for statistical analysis followed by multiple comparison DUNNETs tests. The *p* ≤ 0.05 was considered statistically significant. The results are expressed mean ± standard error (SEM, n = 3). The 50% inhibitory concentration (IC_50_) was calculated for each datum, when a dose–response was observed, using GraphPad Prism 9.0 (GraphPad Software©, San Diego, CA, USA). The other statistical analyses and graphs were performed using the same software [44].

## 3. Results

### 3.1. Chemistry of Isolated 2-(3,4-Dimethoxyphenyl)-3,7-dihydroxy-4H-chromen-4-one

The summary of the overall results is given in Figure 1. The structure of an isolated compound, a natural flavone 2-(3,4-dimethoxyphenyl)-3,7-dihydroxy-4H-chromen-4-one is shown in Figure 1. For convenience, different positions of the compound are labelled with numbers. The ^1^H NMR of 2-(3,4-dimethoxyphenyl)-3,7-dihydroxy-4H-chromen-4-one is provided in Appendix A. The two methoxy groups at positions 17 and 18 gave two distinct singlets (three hydrogen atoms each) at chemical shifts of 3.749 and 3.879. Similarly, the two hydroxyl groups at positions 19 and 20 gave two separate singlets at chemical shifts of 11.996 and 12.124. All the aromatic protons appeared between chemical shift 6 and 8. The protons at positions 8 and 16 appeared as singlets at 6.743 and 6.931. The protons at positions 5, 12 and 13 gave doublets at chemical shifts 7.90, 7.37 and 7.05 with coupling constant values of 179, 1.85 and 3.49 Hz respectively. The proton at position 6 of the flavone gave a doublet at 6.58 ppm with coupling constant values of 1.79 and 3.24 Hz. The ^13^C NMR of 2-(3,4-dimethoxyphenyl)-3,7-dihydroxy-4H-chromen-4-one is provided in Appendix A. In ^13^C NMR spectrum, the compound 2-(3,4-dimethoxyphenyl)-3,7-dihydroxy-4H-chromen-4-one gave signals at 180.52, 160.31, 156.73, 147.91, 146.80, 145.42, 135.51, 128.80, 123.75, 119.52, 113.56, 111.07, 109.61, 108.59, 101.43, 58.25 and 57.15. The MS spectrum of compound 2-(3,4-dimethoxyphenyl)-3,7-dihydroxy-4H-chromen-4-one is provided in the Appendix A. The molecular weight and fragmentation pattern of the compound was 314 (100%), 298 (16%), 271 (18%), 227 (4%), 195 (4%), 157 (20%), 135 (27%), 115 (5%), 85 (18%) and 63 (4%).

### 3.2. Cholinesterase Inhibition Assays

Results of the cholinesterase inhibition assays is summarized in (Table 1). In cholinesterase, our tested flavone exhibited dose-dependent inhibitory activity against acetylcholinesterase (AChE) and butyrylcholinesterase (BChE) enzymes. The observed IC_50_ values of our isolated flavone were 1.37 and 0.95 µM against acetyl and butyrylcholiesterase enzymes, respectively. This activity was compared with the standard galantamine. Galantamine, under the same set of observations, exhibited IC_50_ values of 5.11 and 15.29 µM against acetyl and butyrylcholiesterase enzymes, respectively. Our isolated natural flavone was found to be more potent compared to the standard drug galantamine.

### 3.3. MAO-B Inhibition Study

Our tested sample was found highly effective against MAO-B enzyme as shown in Table 1. The IC_50_ value calculated from the dose-response curve of isolated flavone was 0.14 µM. Standard drug safenamide gave IC_50_ of 0.025 µM.

### 3.4. Anti-Inflammatory Studies

#### 3.4.1. In-Vitro Anti-Inflammatory Studies (COX-1, COX-2 and 5-LOX)

Results of COX-1/2 and 5-LOX inhibitory studies of isolated 2-(3,4-dimethoxyphenyl)-3,7-dihydroxy-4H-chromen-4-one are summarized in Table 2. Our isolated flavone inhibited both COX-1 and 2 enzymes, giving IC_50_ values of 7.09 and 0.38 µM. The observed selectivity index of our compound was 18.70 which shows that our compound will have selective COX-2 inhibition properties, thus protecting stomach problems associated with non-selective inhibitors. In comparison, the respective standard drugs celecoxib (for COX-2) and aspirin (for COX-1) gave IC_50_ values of 0.07 and 32.19 µM, respectively. Similarly, in 5-lipoxygenase assay, our isolated flavone exhibited IC_50_ of 0.84 µM in comparison to the standard montelukast (IC_50_ 4.57 µM).

#### 3.4.2. In-Vivo Anti-Inflammatory Study

Results of the in-vivo anti-inflammatory study at 10 mg/kg dose using carrageenan induced inflammation model are given in Figure 2. During the first hour of the inflammation induction, aspirin exhibited 47.54% inhibition of inflammation and our test flavone reduced inflammation by 45%. During the second hour, aspirin percent edema reduction was 54.63%, whereas our sample showed a 47% decline in paw edema. Likewise, the percent inhibition for aspirin were 53.26 (third h), 56.92 (fourth h), 57.64 (fifth h) and for test compound it was 42% (third h), 49% (fourth h) and 45% at fifth h respectively.

### 3.5. Antioxidant Studies

#### 3.5.1. In-Vitro Antioxidant Studies Using ABTS and DPPH Assays

In ABTS and DPPH anti-radicals’ assays, our tested flavone revealed considerable antioxidant potential (Table 3). The compound gave IC_50_ values of 33.41 and 44.27 µM against ABTS and DPPH free radicals, respectively. The potency of our test compound was compared with positive control acarbose which exhibited IC_50_s of 12.70 and 16.03 µM against ABTS and DPPH free radicals, respectively.

#### 3.5.2. In-Vivo Antioxidant Studies

In-vivo studies revealed that our test compound (2-(3,4-dimethoxyphenyl)-3,7-dihydroxy-4H-chromen-4-one) considerably augmented the level of first line defense antioxidant enzymes. As summarized in Figure 3, treatment with 5 mg/kg and 10 mg/kg has increased SOD levels to 8.52 ± 0.51 UL/mg and 9.58 ± 0.41 UL/mg, respectively, in CCl_4_ treated group which was 6.75 ± 0.17 UL/mg, originally. The SOD blood levels were increased from 0.04 ± 0.02 UL/mg (CCl_4_ treated group) to 0.12 ± 0.02 ^#^ (5 mg/kg) and 0.12 ± 0.32 (10 mg/kg), respectively. Likewise, MDA blood levels were 1.18 ± 0.17 µmol/L in CCl_4_ treated group which was improved to 1.15 ± 0.05 and 1.05 ± 0.03 ^#^ in CCl_4_ treated group supplemented with 5 mg/kg and 10 mg/kg of flavonoid respectively. TP levels were high in CCl_4_ treated group, but significantly changed in treated groups.

### 3.6. Docking Studies

The molecular targets were further studied for their interaction exploration via docking studies [45]. Docking studies were carried out by using Molecular Operating Environment (MOE) software. First, we downloaded the three-dimensional crystal structure of the selected molecular targets from a protein data bank (PDB) [46]. The PDB accession code of the downloaded enzymes are: 1EVE for AChE, 4BDS for BChE, 1CX2 for COX-2, 6N2W for 5-LOX, 2V5Z for MAO-2 and 5I38 for tyrosinase (antioxidant activity). After preparation of the enzymes, the docking protocol was validated by using re-dock method. All the native ligands were docked into the binding site of their respective enzyme. The binding orientation and root mean square deviation (RMSD) were calculated. The docking protocols with a RMSD value less than 2 Å were selected for further studies.

For AChE and BChE, the 3-D/2D interactions are shown in Figure 4. The isolated compound interacts with important catalytic anionic site (CAS) residues Trp84 and Phe330 via π-π interactions in the binding site of AChE. Glu199 established hydrogen bond interactions with hydroxyl group of the compound (Figure 4a). In the binding site of BChE, compound 1 stabilized the ligand-enzyme complex by interacting with catalytic anionic site (CAS) residues Trp82 and Tyr128 via π-π interactions and with catalytic triad residue His438 via hydrogen bond interactions (Figure 4b). These interactions confirm the in-vitro selectivity results as the studied compound showed selectivity towards BChE (SI = 1.44). Moreover, the computed binding energy values also confirm the in-vitro results. The computed binding energy of the compound in the binding sites of AChE and BChE are −6.5536 kcal/mole and −7.2560 kcal/mol, respectively.

Further docking analysis in the binding sites of COX-2 and 5-LOX also confirms the in-vitro results. The studied compound forms hydrogen bond interaction with the deeply located important residues Arg513, Phe518 and Ser353 in the selectivity pocket of COX-2. Ser355 also forms hydrogen bond interactions with oxygen atoms of the ring (Figure 5a). The computed binding energy value for the compound in the binding site of COX-2 (PDB ID = 1CX2) is 8.0712 kcal/mol. In the binding site of 5-LOX, three π-π interactions with Phe359, His432, Trp599 and one hydrogen bond interaction with Arg596 stabilized the ligand enzyme complex (Figure 5b).

In-vitro experimental results showed that the studied compound emerged as an excellent MAO-B inhibitor with IC_50_ value of 0.14 ± 0.01 µM. The activity is due to its strong hydrophobic and hydrogen binding interactions with substrate cavity residues Tyr435, Tyr60, Cys172, respectively (Figure 6a). The computed binding energy value of the compound in the binding site of MAO-B is −9.8195 kcal/mol. For antioxidant activity, the compound was docked into the binding site of tyrosinase (PDB ID = 5I38). The compound coordinated with Cu301 via hydroxyl group oxygen atom and carbonyl group. Phenyl ring is involved in hydrophobic interactions (π-π interactions) with His60, His204 and His208 (Figure 6b).

### 3.7. Pharmacokinetics and ADMET

The physico-chemical and absorption, distribution, metabolism, excretion, and toxicity (ADMET) characteristics of the isolated flavone were deeply investigated and discussed in Table 4. According to Table 4, the isolated compound showed good results within the limit for lipophilicity, insolubility, size, instauration, polarity, and flexibility. The oral bioavailability chart of the isolated compound is mentioned in Figure 7A. Among all six factors, only the instauration was out of the limit.

HIA and CNS absorption are essential parameters checked for every drug before its entry for drug formulation in the pharmaceutical or clinical trials field [47]. The blood–brain barrier penetration is essential as the compounds that act on the central nervous system (CNS) must cross through the blood–brain barrier, and the inactive compounds on the CNS should not intersect to avoid adverse effects on the CNS [48]. As mentioned in Table 4, the isolated flavone displayed a high gastrointestinal absorption (HIA) with less BBB permeability, indicating that isolated flavone shows low occurrence for adverse CNS effects. 

Figure 7B shows the BOILED-EGG curve. The BBB penetration and GI absorption (HIA) of the substances may be predicted by this method. There are two areas: one for the GI absorption zone (HIA) and the other for BBB penetration (yolk). Neither GI absorption nor BBB penetration is indicated if any component is found in the gray zone. Furthermore, isolated flavone did not show that it is a P-gp substrate and it is not sensitive to the efflux mechanism of P-gp, which is used by many cancers’ cell lines to develop resistance to drugs Figure 7B.

Isolated flavone has shown 90% absorption orally. It has poor penetration into the CNS and cannot cross the blood–brain barrier (BBB). The isolated flavone is an inhibitor of CYP2C19 and CYP1A2 metabolic enzymes. It has total clearance of 0.691 as mentioned in Table 4. The high LD_50_ value presented the drug as safe to use as well. There was no hepatotoxicity indicated for the isolated flavone, which indicates its safety zone. Furthermore, the isolated flavone was also a non-inhibitor for hERG I. In addition, the isolated flavone did not show any environmental toxicity was included, and the isolated compound was also a non-skin sensitizer.

#### Cardiac Toxicity

The FDA requires that every biomolecule should be tested for hERG safety before it may be used as a therapeutic candidate. The hERG blockage has been connected to deadly cardiac arrhythmias. Using pred-hERG results to predict cardiac toxicity, the likelihood map for isolated flavone is shown here (Figure 8). Attributions to hERG blockage, both positive and negative, are shown in the figure. Increasing the number of contour lines and the intensity of the green color shows that an atom or fragment has made a more positive contribution to the hERG blockage. With a 60% confidence level, the pred-hERG projected that isolated flavone would be non-cardiotoxic. The findings have revealed that our isolated flavone is safe for cardiovascular toxicity.

### 3.8. Skin Sensitization

The isolated flavone was evaluated for its skin sensitization in the in-vitro human cell line activation test (h-CLAT) OECD442E and in-vitro KeratinoSens OECD429. The cellular response of isolated flavone was evaluated against dendritic cells for the induction of inflammatory cytokines and the mobilization of dendritic cells. The h-CLAT- OECD442D results have shown that isolated flavone was conducted as a non-sensitizer with a confidence score of 59.6% (Figure 9A). Furthermore, in-vitro KeratinoSens OECD442D evaluation of isolated flavone was evaluated for activation of inflammatory cytokines and inducing cytoprotective genes response OECD442D. The results have shown that the isolated flavone was conducted as a non-sensitizer with a highest confidence score of 91.6% (Figure 9B). From cellular response, it also proved that isolated flavone is a non-sensitizer for histocompatibility complex, represented by dendritic cells.

## 4. Discussion

Alzheimer’s disease (AD) is among the highly prevalent neurological disorders which drastically effects patients’ quality of life and is a posing a major burden on the healthcare system [49]. Unfortunately, the drug discovery to cure the disease is very limited and only five drugs are clinically approved, which are also associated with limited efficacy [50]. Thus, there is dire need for the discovery of more effective and safe drugs, especially from natural products [51]. Polypenols is a group of highly potent secondary metabolites which are extensively reported for their pre-clinical efficacy against various pathological targets of AD [17]. Hereon, we are reporting the detailed study on the efficacy of a potent flavone 2-(3,4-dimethoxyphenyl)-3,7-dihydroxy-4H-chromen-4-one from *Notholirion thomsonianum* against various pathological targets including free radicals, inflammation and cholinesterases, using in-vitro, in vivo and molecular docking approaches. 

Acetylcholinesterase (ACh) is an important neurotransmitter implicated in the transmission of impulses across the synapse towards the effecter organ. Once released from the pre-synaptic nerve terminal, ACh interacts with its receptors, causes activation of these receptors and subsequently an impulse of considerable intensity is transmitted towards the post-synaptic neuron [42]. The action of ACh is terminated by the action of these esterases, which metabolize and recycle them inside a pre-synaptic nerve terminal [52]. In AD, there is selective deterioration cholinergic neurons, and as a result there is a considerable deficiency of ACh, thus causing difficulty in the consolidation of memory and routine life activities. Among the therapeutic options is to inhibit the esterases which metabolize the ACh, so that the ACh released will remain for a prolonged time at the nerve terminal and a dmore intense response can be produce [53]. Among the currently discovered anti-AD drugs, clinicians mostly rely on these cholinesterase inhibitors as they are the only effective therapeutic strategy until now. Plant-based cholinesterases are more effective and the discovery of galantamine, as well as rivastigmine, further strengthens this notion [45]. Several plant-derived polyphenols have previously been reported to exhibit cholinesterase inhibitory potential [54]. Cholinesterase inhibitory potential of quercetin, kaempferol and rutin isolated from the fruit of *Prunus persica* is reported [55]. Zhou y et al., reported the anti-amyloid and cholinesterase inhibitory potential of phenolic acids from *Salvia miltiorrhiza* [56]. Furthermore, polyphenols also exhibit free radical scavenging potential, and thus reduce oxidative stress. Subsequently, they reduce free radical-mediated inflammatory processes and neuro-degeneration [57,58]. Keeping in view the significance of flavonoids as neuroprotective agents, and to find more effective cholinesterase inhibitors, we isolated and tested the current flavonoid/flavone from a highly potential plant. The compound appears to be effective on more than one pathological target, as it inhibits cholinesterases, scavenges free radicals and exhibit santi-inflammatory potential.

MAO-B is another enzyme which plays its physiological role in the recycling of an important neurotransmitter, dopamine. Thus, inhibitors of the dopamine metabolizing enzymes are known to play protective roles against Parkinson’s disease, oxidative stress and AD [59]. In AD patients, it has been reported that there is a three-fold elevation in the level of the MAO-B enzyme which leads to unwanted outcomes including excessive production of free radicals [60]. Subsequently, MAO-B selective inhibition, especially from plant-based origins, are in clinical trials for the effective management of AD [61]. The MAO-B inhibitory potential of polyphenols is well established. The inhibitory potential of curcumin, its metabolite tetrahydrocurcumin and ellagic acid has previously been reported [62,63]. Berry anthocyanins [64], *Annurca apple* polyphenols [65] and *Uncaria rhynchopylla* is previously reported [66]. Thus, our test compound, owing to its considerable efficacy against MAO-B, AChE/BChE and inflammatory enzymes, is of high significance as a multi-target agent against AD.

Cyclooxygenases are enzymes which, beside their gastroprotective role, are implicated in the production of the prostaglandins involved in inflammation, pain and pyrexia [67]. Neuro-inflammation is among the key pathological aspects of the degenerative disorders including AD [68]. Administration of anti-inflammatory agents is of considerable significance to reduce Aβ and the free radical-induced inflammatory process in the brain. Natural flavonoids are of particular interest in relieving neuro-inflammation [69]. Plant-derived polyphenols exhibit considerable anti-inflammatory potential [70]. Polyphenols mediate their anti-inflammatory potential via inhibition of NF-kB-induced cytokines production in AD [71]. Free radicals being implicated in inflammatory processes is an important target in AD. Polyphenols-rich extracts from *Stachys officinalis* and *Impatiens noli-tangere* exhibit both antioxidant and anti-inflammatory potential [72]. In our study, our test sample showed the nonselective inhibition of COX-1 and COX-2 and the anti-inflammatory activity was comparable with control drugs. Additionally, the production of inflammatory eicosanoids by cyclooxygenases including 5-LOX is of considerable significance in AD. Research studies indicate that these eicosanoids produced in the brain cause neuro-inflammation and thus 5-LOX inhibitors coupled with other therapeutics are very effective in AD, stroke and ischemia [73,74]. Our test compound showed dose-dependent anti-inflammatory potential, which suggests that our compound is effective on more than one target.

Medicinal chemistry researchers are in constant search of new drug molecules or drug building blocks from both natural and synthetic sources [75,76,77]. Molecular docking has been one of the convenient approaches to find out the binding interaction of a compound in the binding pocket of the target protein [78]. The molecular docking approach correlates the in-vitro studies with in-silico studies. In this study, we have docked our isolated flavone with the tested in-vitro targets, and we discovered binding interactions in all in-vitro targets.

Free radicals are generated during metabolic processes in the body and are subsequently neutralized by natural antioxidant system like catalases and hydro-peroxidases [79,80]. However, in case of excessive free radical production or the body’s inability to scavenge the free radicals effectively, supplementation of exogenous antioxidants is extremely necessary [81]. In AD, Aβ proteins are deposited in the brain, which is considered as a mitochondrial poison. After production, it readily attacks lipids, membranes, proteins and nerve cells, causing genetic mutations, cells deterioration and the disruption of energy production by mitochondria [82]. Subsequently, supplementation of antioxidants is an integral part of the drug combinations for AD patients. Natural products and their derived compounds have received considerable attention as potential multi-target agents. Of particular interest is polyphenols which, being part of natural products and nutraceuticals, can delay the aging process and can act on multiple targets of AD. Numerous flavonoids are reported to inhibit cholinesterases, BACE1, scavenge free radicals and act as anti-inflammatory agents [83]. In the current study, our test compound showed efficacy on multiple targets and thus acts as a multi-target lead compound. However, further detailed studies regarding the bioavailability and in-vivo efficacy are required.

## 5. Conclusions

In this research, we have isolated 2-(3,4-dimethoxyphenyl)-3,7-dihydroxy-4H-chromen- 4-one from *N. thomsonianum*. The structure was confirmed with spectral analyses. The compound was tested against various in-vitro targets such as AChE, BChE, MAO-B, COX-1, COX-2, 5-LOX, ABTS and DPPH. In all the in-vitro assays, it was found to be a potent inhibitor. We also extended our research to experimental animals. The carrageenan-induced inflammation was cured with our isolated flavone in experimental mice. Similarly, the isolated flavone at a dose of 5 and 10 mg/kg considerably enhanced the level of first line defense antioxidant enzymes and was observed to be a potent antioxidant. Overall, we can conclude that our isolated 2-(3,4-dimethoxyphenyl)-3,7-dihydroxy-4H-chromen-4-one from *N. thomsonianum* has been effectively employed in the management of oxidative stress-related neurological disorders and inflammation. Further research on the same isolated compound to embark on its further potential roles is in progress in our research laboratory. The compound was docked into the target proteins of the respective activities and the binding energies confirmed the potency of our compound.

## Data Availability

The whole data is available within the manuscript and Appendix A.

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
