# Peer review of "In-Vitro, In-Vivo, Molecular Docking and ADMET Studies of 2-Substituted 3,7-Dihydroxy-4H-chromen-4-one for Oxidative Stress, Inflammation and Alzheimer’s Disease"

_metabolites, 2022, doi:10.3390/metabo12111055_

Round 1

Reviewer 1 Report

General Comment:

The manuscript is well written. The authors investigated “In vitro, in-vivo, molecular docking and ADMET studies of 2-substituted 3,7-dihydroxy-4H-chromen-4-one for neuroinflammation”. The study is interesting and adds to the existing body of knowledge. Some errors need clarification and revisions. The manuscript should undergo English editing. 

Detail’s comment:

1.     Page 1, Line 2-3: Please capitalized each word in the title based on the journal’ format

2.     Page 1, Line 31: carrageenan-induced

3.     Page 1, Line 44: Please add space/press “tab” before starting a paragraph. Please standardize all in the manuscript. 

4.     Page 2, Line 91: Please add numbering for the sub-title in the materials and methods sections, and please capitalized each word in the sub-title. 

5.     Page 3, Line 98-102: Please explain in detail plant extraction (weight of powder used, extraction ratio, extraction temperature).

6.     Page 3, Line 103: Please add numbering for sub-title and capitalized each word

7.     Page 3, Line 118-119, 127: Please add numbering for sub-title and capitalized each word

8.     Page 3, Line 128-146: Please change all the sentences in the paragraph to passive sentences.

9.     Page 3, Line 138: Please add numbering for sub-title and capitalized each word

10.  Page 4, Line 147: Please add numbering for sub-title and capitalized each word

11.  Page 4, Line 170-171, 178: Please add numbering for sub-title and capitalized each word

12.  Page 5, Line 187-191: Please change all the sentences in the paragraph to passive sentences.

13.  Page 5, Line 192-193, 207: Please add numbering for sub-title and capitalized each word

14.  Page 5, Line 210-211: Please check the sentences

15.  Page 5, Line 220-228: Please provide the age of mice, temperature, and light and dark cycles (hours) of animal handling. Please also add a reference number for animal ethics. Please also add the total number of rats used in the current study. 

16.  Page 6, Line 229, 243, 259, 262: Please add numbering for sub-title and capitalized each word

17.  Page 6, Line 240-242: Please add references for the sentence. 

18.  Page 6, Line 252: Please state the volume of blood used

19.  Page 6, Line 252: Please standardize the word “minutes” to “min” in the manuscript.

20.  Page 7, Line 271, 284, 291, 300, 310: Please add numbering for sub-title and capitalized each word

21.  Page 7, Line 282: Please add references for the sentence

22.  Page 7, Line 287-288: (sod. hydroxide with thiobarbituric acid, 2ml). Please rewrite this. 

23.  Page 7, Line 292: Please add references for the sentence.

24.  Page 7, Line 297: “normal laboratory temperature”. Please add “room temperature” or exact value.

25.  Page 8, Line 317, 322, 326: Please add numbering for sub-title and capitalized each word

26.  Page 8, Line 326-328: Please rewrite the paragraph for statistical analysis. Please provide software (version, company name) used for analysis and significant value for analysis. 

27.  Page 10, Line 393-394: Please add significant value for statistical analysis. Please put the mark (# and *) in which group is compared for. 

28.  Page 11, Line 405-406: Put the compound name

29.  Page 11, Line 415-418: Please labeled the figure correctly. (A) SOD, (B) CAT, (C) MDA, (D) TP. 

30.  Page 14, Line 479: Please labeled the figures for (a) and (b).

31.  Page 15, Line 502: LD50 

32.  Page 17, Line 534: Please labeled the figures for (a) and (b).

33.  Page 19-22, References: Please check the references styles and change the format as follows: https://www.mdpi.com/journal/metabolites/instructions#references

Reviewer 2 Report

The present paper, entitled "In-vitro, in-vivo, molecular docking and ADMET studies of 2-substituted 3,7-di- 2 hydroxy-4H-chromen-4-one for neuroinflammation, brings new natural compounds in attention for their antiinflamatory and neuroprotective effects". Their antioxidant properties were a checked, oxidative stress being one of the most important inflammatory trigger.

Introduction

Please refer to this pathogenetic loop  (oxidative stress as trigger for inflammation) to explain, and to make connection between your assessed parameters (antioxidants) and neuroinflammation.

Material and Methods

Please provide the number of Ethical Committee approvement.

Line 225-228 - please describe with more details the acute toxicity experiment test.

Please provide a scheme with all your experiments, to be more clear for reading. As you wrote each experiment is difficult to read rapid your methods for each of them, because they are complex.

line 296 - replace Foli ciocalteu with Folin Ciocalteu.

Because of your title and your study aim, please describe more details (separately) about the BBB permeability assessment.

Discussion

Please indicate more studies, similar with your research, and compare your results with other similar results. The discussions are to general and should offer more details about other similar studies in order to observe the differences and, eventually the novelty/improvment of anti neuroinflammation efficacy of their tested compounds.  

Round 2

Reviewer 2 Report

The manuscript entitled "In-vivo, Molecular Docking and ADMET Studies of 2-substituted 3,7-di- 2 hydroxy-4H-chromen-4-one for Neuroinflammation"  brings a new detail about antio-xidant properties of plants extrct

The following corrections have to be made:

Please change the title; the "neuroinflammation" word  is unproper in your title since your experimental method is about arrageenan induced inflammation. This is a clasic method of inflammation, not an experimantal neuroinflammation. There is no direct conection between your study results and Alzheimer's disease. You can mention in discussion the pathophysiological mechanisms of Alzheimer's disease and your study results. Otherwise is only an indirect assumption of the efficacy of your results in Alzheimer's disease. Eventually you can mention your studied molecules as a potential treatment.

Introduction

Please write the aim of this study in the end of introduction. You only wrote your previous results and research. Specify clear what is the novelty of your present work.

Material and methods

Line 257 - Since the other methods are detailed described, please make a short description of carrgeenan induced inflammation. You only offered reference.  I consider the experimental induced inflammation by carrageenan a central point of your study.

Conclusions

L:ine 677-678 - reformulate the sentence - specify more clear how your study molecules were "potent" (offer details please).
